# A Gap between Relaxation of Government Quarantine Policy and Perceptions of COVID-19 among the General Public in Sports: Focusing on Vaccination Status

**DOI:** 10.3390/ijerph19074267

**Published:** 2022-04-02

**Authors:** Mun-Gyu Jun, Kyung-Rok Oh, Chulhwan Choi

**Affiliations:** 1Department of Coaching, College of Physical Education, Kyung Hee University, Seocheon-dong 1, Giheung-gu, Yongin-si 17104, Korea; mkrollcage@khu.ac.kr; 2Department of Physical Education, Gachon University, Sujeong-gu, Seongnam-si 13120, Korea

**Keywords:** social distancing, COVID-19, vaccination, sport participation

## Abstract

Although an increasing number of people are getting vaccinated for COVID-19 and quarantine policies are easing owing to fatigue from high-intensity social distancing, people’s fear remains. This study attempted to determine the appropriateness of quarantine policies that are gradually easing by comparing and analyzing sports participation and respiratory infection perception recognized by sports participants according to vaccination status. Data were collected from 302 ordinary Korean citizens aged 20 or older for three months from November 2021 in the Republic of Korea. From the survey respondents, data on the main factors of this study included (a) demographic information, (b) vaccination, (c) loyalty in sports, (d) behavioral intention to participate in sports, (e) infection anxiety from others, and (f) risk perception of COVID-19. As a result, the survey respondents, subdivided into an unvaccinated group (Group 1) and a vaccinated group (Group 2), derived statistically significant results on sports participation and respiratory infection perception. Specifically, survey participants who had completed all secondary vaccinations showed a relatively higher (a) loyalty in sport (*M* = 3.789), (b) behavioral intention for participation in sport (*M* = 4.056), and (c) infection anxiety from others (*M* = 3.548), but showed a relatively lower (a) risk perception of COVID-19 (sensitivity) (*M* = 3.494). The results of this study could be utilized as valuable data to minimize the gap between the relaxation of government quarantine policy and perceptions of COVID-19 among the general public in sports, which have not yet been clarified.

## 1. Introduction

In December 2019, COVID-19, a new type of virus, appeared worldwide [1]. The World Health Organization (WHO) [2] declared the novel coronavirus (COVID-19) outbreak a global pandemic on March 11. As of 5 February 2022, 385,604,041 people had been confirmed to be infected worldwide, of which 5,698,892 died [3]. By 7 February 2022, 1,044,963 people were confirmed to have COVID-19, and the death toll had reached 6886 in the Republic of Korea [4]. However, since these numbers may not be accurate, there may actually be many more infections and deaths. As a result, anxiety over the spread of infection has made it rare to see people without masks anywhere, and vaccination is an important factor in preventing the formation of immunity in groups and the spread of infection in individuals and communities [5,6,7,8]. The government has recorded a vaccination rate of more than 80% [9] as of 7 February 2022, because of the massive promotion of vaccines and recommendations for vaccination. In addition, rather than expecting a complete end to COVID-19, the government prepared for step-by-step daily recovery to coexist with COVID-19 by easing the social distancing system (e.g., prohibition of gathering, lockdown, and business hour restriction). However, not long after the government implemented the relaxed quarantine system (i.e., shortened quarantine period for confirmed patients from 14 days to 7 days), high-intensity social distancing (e.g., lockdown after 8 pm, closing of school, and working from home) was re-implemented without any difference between vaccinated and unvaccinated from 18 December 2021, owing to the explosion of confirmed COVID-19 cases and the emergence of new mutations, such as Omicron (SARS-CoV-2 Variant, B.1.1.529), following previous mutant viruses (i.e., Alpha, Beta, Gamma, and Delta) [10].

In addition to the global spread of COVID-19, the nationwide spread of COVID-19 has caused various changes in daily life [11]. First, individual psychological fear and anxiety about COVID-19 infection occur [12], and the government’s social distancing policy not only restricts physical and daily activities, but also restricts family and religious activities [13]. In addition, the environment in which the use of various sports facilities nationwide has been restricted due to COVID-19 can negatively affect mental health and lower happiness levels, as continuously participating in sports promotes mental health and happiness through exercise [12]. In particular, participants in sports who had maintained their physical health through physical activities such as daily sports before COVID-19 often stopped exercising because of social distancing, quarantine guidelines, and restrictions on the use of sports facilities [14]. Since the outbreak of the coronavirus, researchers have been conducting various studies emphasizing the importance of factors such as physical activity participation, mental health, well-being, and self-quarantine [15,16,17]. Reports state that those participating in physical activities such as sports for all discover their active appearance and achieve new challenges and efforts on their own, which positively affect life satisfaction [18,19,20].

After two years, the new virus is still ongoing, but exhausted people prepare to coexist with COVID-19, although they risk falling ill with it. This is because the spread of vaccines worldwide has become smooth, and the Omicron mutation, which is highly contagious but has a lower fatality rate than other mutations, is dominant. Infectious disease experts have argued that it is impossible to “end” COVID-19 in the form of complete eradication on earth and have agreed that humanity should prepare for a “coexistence” that lowers the risk of COVID-19 to a manageable level. However, there are still many processes to overcome, and it is never an unsafe situation. This is because the COVID-19 situation, which showed a lull for a while owing to active participation in social distancing, vaccination, and efforts by quarantine and medical authorities, has shown fierce momentum and proved its presence with stronger explosive power [21].

Loyalty, which can be seen as a rather abstract concept, might have originated from the concept of organizational identification, that is, the concept of identification between teams and fans in terms of sports. According to Kelman [22,23], the concept of loyalty has been defined as maintaining self-defined relaxation in a positive direction toward what individuals find attractive. In addition, loyalty in the sports industry has been regarded as an important factor closely related to consumers’ actual purchasing behavior [24]. In addition, it can be said that an unspecified number of people with various interests directly influence active behavior according to their degree of loyalty [25]. In terms of sports marketing, loyalty, as well as satisfying the needs of people, is the most important factor for revitalizing the sports industry [26]. Recently, in sports academia, active research has been conducted on how loyalty is influenced or reflected in people with common interests in special situations where non-face-to-face activities are prioritized over face-to-face activities owing to COVID-19.

Consideration of participation intention precedes intention. Intention means that each individual’s beliefs are transferred to concrete actions toward the planned future [27]. The term “intention” should necessarily be premised on all participatory actions of each individual. Intention is regarded as having the greatest influence in determining an action and is greatly influenced by each person’s personal socio-cultural situation and psychological factors [28]. Similarly, the intention to participate in sports also indicates the decision of what situation to participate in and the state of final resolution, accordingly [29]. Through participation in comprehensive activities, including physical, emotional, psychological, and social activities, individuals can draw satisfaction from a pursuit of happiness and value [30], which reflects the positive attitudes and values toward sports that draw continued participation [31].

The prevalence of new respiratory infectious diseases, such as COVID-19, has increased infection anxiety. In fact, 55.8% of people in the Republic of Korea felt somewhat or seriously anxious or depressed owing to the spread of COVID-19 [32]. Considering previous studies on hospital workers related to respiratory infectious disease infection, the assumption is that the level of negative emotions, such as the anxiety, of nursing college students participating in practice in situations similar to hospital workers will also be high in the event of severe acute respiratory syndrome virus (SARS). In addition, there have been studies on influenza incidence factors [33], swine flu prevention behavior [34], knowledge of swine flu, and perceived threats [35], but few studies have been conducted on the general public participating in sports activities related to respiratory infectious diseases.

Respiratory infectious diseases are rapidly or chronically transmitted from person to person by pathogens, causing great confusion in society and a very high risk of infection and spread [36]. To prevent and manage infectious diseases, legal infectious diseases are divided into levels 1 to 4, and first-class infectious diseases have a high fatality rate and are highly likely to occur in groups [37]. In this study, it has been defined as standard for infectious diseases of the respiratory tract to be among infectious diseases corresponding to the first level of legal infectious diseases. Recent studies on infectious diseases include those on the risk perception of infectious diseases [38] and sports activities for respiratory infectious diseases [39].

The government may find it difficult to continue maintaining a strong quarantine system when people’s fatigue from quarantine measures has reached its peak. However, in various overseas cases, quarantine regulations have been eased, and the spread has significantly increased [40]. People also expect quarantine policies to be relieved; however, they are confused by the realistic fear of the re-proliferation of COVID-19. Therefore, this study aims to identify the current situation and present future strategies by analyzing the gap between the government’s relaxation of quarantine policies and COVID-19 perceived by people in terms of their sports participation. Specifically, this study set the vaccination status, which currently has the most significant impact on people’s lives, as an important research criterion and analyzed people’s perception of sports participation and COVID-19 through comparative studies.

## 2. Materials and Methods

### 2.1. Data Collection Procedure

Despite the recent development of and inoculation with vaccines, COVID-19 is spreading owing to the development of mutant viruses. In this situation, this study attempted to determine the appropriateness of quarantine policies that are gradually easing by comparing and analyzing sports participation (loyalty in sports and behavioral intention to participate in sports) and respiratory infections (infection anxiety from others and risk perception of COVID-19) perceived by sports participants according to vaccination status. To achieve this research purpose, among Korean adults aged 20 years or older in the Republic of Korea, sports participants who were enjoying leisure sports for more than a year since the COVID-19 outbreak formed the inclusion criteria. The data collection process was conducted in two universities located in the Republic of Korea from 5 November 2021 to 9 January 2022. Moreover, the procedure adopted a quantitative research design; a convenience sampling via intercept survey technique was applied to comply with the qualifications of accurate survey participants and proceed with data collection. Both online and offline survey methods were adopted at the level of compliance with the government quarantine policy. Google’s survey platform was used for online surveys. The survey respondents read a brief explanation of the purpose of this study and completed the survey according to their choice (online/offline) using a self-administration method based on voluntary participation. A total of 350 questionnaires were distributed, and 309 were returned (approximately 88.3% response rate). After excluding seven incomplete questionnaires, 302 were finally used for the analysis of this study. The sample size met the statistical criteria (*n* = 302), estimated through the G*power program for a multivariate analysis of variance (*F* test), including four variables with two groups.

The survey participants’ basic demographic information (e.g., gender and age) was collected, and additional surveys related to leisure sports (e.g., the number of participants and favorite events) were also conducted. In addition, the questionnaire was preceded by a question (What is your vaccination status?), which is a key independent variable in the comparative analysis of this study. To answer this question, survey respondents had to choose one of the following: (a) unvaccinated, (b) primary vaccination completed, or (c) secondary vaccination completed. Based on the government’s quarantine guidelines, only those who had completed all secondary inoculations were classified as inoculators. Specifically, the survey participants were subdivided into an unvaccinated group (Group 1) and a vaccinated group (Group 2). Detailed descriptive statistics of the survey respondents are reported in Table 1.

### 2.2. Instruments

To secure content validity and improve the readability of the scale applied in this study, the researchers sent the instruments to a panel of experts, including five faculty members specializing in sports management or physical education. Feedback from the panel of experts was assessed, and minor modifications were made as follows. This study modified the instruments developed in James’s study [41] to measure the loyalty to sports of the survey respondents. Until recently, loyalty scales have been used in previous studies, such as Lee [42] and Lee [43]. The loyalty scale consisted of three questions as a single factor. Next, the instrument used to measure the behavioral intention to participate in sports was a factor re-established by Kim [44] based on the items developed by Park [45] and applied according to the purpose of this study. The behavioral intention to participate in sports factor consisted of a total of three questions with a single factor. In addition, the tool used to measure infection anxiety from others was a questionnaire modified by Kang [46] based on the questionnaire developed by Park [47]. The instrument was applied, which meant that the higher the score, the higher the degree of infection anxiety. Finally, to measure the respondents’ risk perception of COVID-19, an instrument established by Lee [48] based on the risk perception factor of Kang and Kim [49] for acute respiratory infectious diseases was applied to this study. All original instruments already showed satisfactory psychometric information in previous studies as follows: (a) loyalty to sport, (*α* = 0.89), (b) behavioral intention for participation in sport, (*α* = 0.63), (c) infection anxiety from others, (*α* = 0.91), and (d) risk perception of COVID-19, (sensitivity, *α* = 0.74 and seriousness, *α* = 0.85). Nevertheless, this study tested the validity and reliability of all instruments again. The questionnaire items used in this study consisted of a 5-point Likert scale ranging from 1 (“not at all”) to 5 (“very much”).

## 3. Results

### 3.1. Scale Validity and Reliability

The survey scales utilized in this study have already been tested for satisfactory validity and reliability in various previous studies on the sports industry and respiratory infectious diseases. However, as the scales were revised and supplemented based on the research topic and purpose of this study, a confirmatory factor analysis (CFA) was conducted with all five variables. All statistical results for scale validity in this study met the statistical standards. Additionally, the goodness-of-fit test showed satisfactory results (CMIN = 223.840, DF = 94, CMIN/DF = 2.381, NFI = 0.938, CFI = 0.963, RMSEA = 0.068).

Next, the reliability of the survey items as factors was analyzed by applying Cronbach’s alpha, and a satisfactory statistical cutoff value of 0.70 [50] was applied: loyalty to sport, *α* = 0.817; behavioral intention for participation in sport, *α* = 0.869; infection anxiety from others, *α* = 0.889; risk perception of COVID-19 (sensitivity), *α* = 0.862; and risk perception of COVID-19 (seriousness), *α* = 0.803. Thus, all the instruments exceeded satisfactory statistical reliability. All the detailed results of the CFA and Cronbach’s alpha are shown in Table 2.

### 3.2. Multivariate Analysis of Variance (MANOVA)

A one-way MANOVA for the comparative study was implemented to verify the differences in loyalty to sports, behavioral intention to participate in sports, infection anxiety from others, and risk perception of COVID-19 in sports depending on vaccination status in the era of COVID-19. First of all, three statistical assumptions (multivariate normality, independence, and outliers) were ensured. Moreover, to check another assumption of equal variance, the homogeneity of covariance was tested (Box’s *M* = 137.958, *F* = 8.988, *p <* 0.001). Statistically significant differences between the two groups were found (Wilks’ Lambda = 0.575, *F* = 43.669, *p* < 0.001). As shown in Table 3, statistically significant differences between groups were found in four dependent variables: (a) loyalty to sports, (b) behavioral intention to participate in sports, (c) infection anxiety from others, and (d) risk perception of COVID-19 (sensitivity). However, no statistically significant differences were revealed in seriousness, a sub-factor of risk perception of COVID-19. All mean scores of the five dependent variables between the groups according to vaccination status are reported in Table 4.

## 4. Discussion

Although all countries are actively implementing quarantine policies, the spread of COVID-19 is still limiting people’s daily lives and threatening global public health. The spread of the virus, which had reduced owing to the development of the vaccine, has taken another turn as a result of the outbreak of the mutated virus, which is now out of control. The number of people infected with SARS-CoV-2 is still rising out of control; therefore, it is essential to analyze the consistency of government policy with people’s perception of the virus. In terms of public health, one of the biggest restrictions people have experienced in their daily lives has been on their participation in sports activities owing to quarantine rules, and the biggest criterion for these restrictions is the individual’s vaccination status under the current circumstances. This is because, according to the quarantine rules announced by the government, people’s vaccination status serves as an individual’s identity card. Therefore, this study compared and analyzed the differences in (a) loyalty to sports, (b) behavioral intention to participate in sports, (c) infection anxiety from others, and (d) risk perception of COVID-19 in the current situation where quarantine policies are easing.

First, among the results of this study, statistically significant differences in the factor of loyalty to sports were found according to vaccination status. Specifically, survey participants who had completed all secondary vaccinations showed a relatively high loyalty to participate in physical activities. The term “loyalty” has been defined as a measure of continuous participation and interest [25,42,43]; therefore, it should be considered an important factor in the COVID-19 outbreak because loyalty expresses a willingness to continue participating in sports even with the government’s quarantine rules and social distancing. In this study, participants who completed all vaccinations showed a high loyalty to sports, actively accepted the government’s quarantine policy, and showed a willingness to continuously participate in physical activities. This is a positive psychological response to the vaccine effect, which is an active practice not only for the government’s policy, but also for the recovery of daily life. This could be explained by the fact that individuals’ perceptions of the virus coincided with the easing of the quarantine policy analyzed in this study. Considering the results of this study and the current situation, in which more than 80% of the people have been vaccinated, a quarantine policy that focuses on the easing of restrictions while preparing for coexistence with COVID-19 may be positive. Given the fact that COVID-19 is perceived as a threatening disease [51], most citizens have made efforts to protect themselves from the threat of coronavirus. In the sports industry, it is necessary to prepare for these changes and establish future plans and strategies.

Statistically significant results were derived from the factor of the behavioral intention to participate in sports. The concept of “intention” could be analyzed slightly differently from the aforementioned “loyalty” because the “intention” implies a slightly more active individual will. Specifically, since the term means that each individual’s beliefs are transferred to specific actions [52], behavioral intention to participate in sports is accompanied by actual actions, rather than loyalty. The results of this study also showed high results for the factor of the intention to participate in sports in the vaccination group, who showed high loyalty. This indicated that not only did the vaccinated group have psychological loyalty, but also strongly expressed a willingness to actually participate in sports. This represents the government’s quarantine policy, which provides many benefits to vaccinated individuals. Non-vaccinated people choose not to be vaccinated, even while recognizing that their personal lives will be restricted. According to a study of Lin, Tu, Beitsch [53], vaccine hesitancy is widespread due to the perceived risk, concerns over vaccine safety and effectiveness, doubts toward the expedited development/approval process, and so on. This places a higher value on free will than on personal life, and there is a possibility that people are participating in home training or outdoor physical activities. In terms of the gap between the quarantine policy focused on in this study and people’s perceptions, it is possible to assume that the easing quarantine policy, similar to the result of the loyalty factor, could be supported.

In addition, the results of the vaccinated group were high in the factor of infection anxiety from others, a factor related to the threat of COVID-19. These results contradict the expected results. Even if vaccination is aimed at lowering the risk of infection from the virus itself and preventing it from serious post-infection outcomes [54], the vaccinated group showed relatively higher results in the factor related to the threat of COVID-19 than the group that did not complete the vaccination. These results suggest that those who have completed vaccination are more sensitive to infection from others than to the fear of COVID-19 itself. This analysis could be linked to the results of the risk perception of COVID-19, another factor applied in this study among the factors related to the threat of COVID-19. This is because the risk perception of COVID-19, a factor in analyzing the threat of the coronavirus, showed high results in the groups with incomplete vaccination. As such, it could be interpreted that the reason for vaccination is to prevent infection from others. This is consistent with results from a study by Burke, Masters, and Massey [55] that trust in the persistent effectiveness of the vaccines reduces anxiety about infection. In contrast, it could be interpreted that the unvaccinated group had a higher risk perception of the virus itself than of infection with others. Nevertheless, a change in perception may occur if it is time to coexist with COVID-19 in the future. Statistical results from the two factors of the risk perception of COVID-19 and infection from others have been derived in reverse, which is progressing rapidly, and the perceptions of vaccines should change more positively in current situations where quarantine is likely to ease and there is a return to normal life, including sports participation.

## 5. Conclusions

At a time when risks and uncertainties have become commonplace owing to the prolonged spread of COVID-19, the vaccination rate continues to increase, but the number of confirmed cases is also increasing exponentially. This study attempted to determine the appropriateness of quarantine policies that are gradually easing by comparing and analyzing sports participation (loyalty and participation intention) and respiratory infections (infection anxiety and risk perception) recognized by sports participants according to vaccination. Although this study found significant results, there are also research limitations.

First, the age of the survey participants was not adopted as an important variable despite the presence of various variables, such as fear, anxiety, and risk perception of the COVID-19 virus, which can be fatal to elderly sports participants. In this way, given that the segmentation of survey participants was limited, it would be essential to analyze sports participants of various groups by applying additional potential variables (e.g., age and sex) against COVID-19 in future studies.

In addition, other various sports were not included in the survey because of restrictions on the use of many sports facilities owing to social distancing. Since COVID-19 is likely to coexist with daily life rather than be completely extinguished in the future, it is necessary to analyze the effect of COVID-19 on the sports industry from a more expanded perspective. Therefore, a research design that considers more diverse variables and situations should be required in future research.

## Figures and Tables

**Table 1 ijerph-19-04267-t001:** Descriptive statistics for survey respondents.

		Group 1 (Unvaccinated) 89 (29.47%)	Group 2 (Vaccinated) 213 (70.53%)
Gender	Male	59 (66.3%)	155 (72.8%)
	Female	30 (33.7%)	58 (27.2%)
Age	20 s	26 (29.2%)	138 (64.8%)
	30 s	25 (28.1%)	56 (26.3%)
	40 s	21 (23.6%)	9 (4.2%)
	50 s	12 (13.5%)	7 (3.3%)
	Over 60 s	5 (5.6%)	3 (1.4%)
Vaccination status	Unvaccinated	23 (25.8%)	-
	Primary vaccination completed	66 (74.2%)	-
	Secondary vaccination completed	-	213 (100.0%)
Frequency of participation in sports per week	Once	37 (41.6%)	66 (31.0%)
	2–3 days	37 (41.6%)	43 (20.2%)
	More than 4 days	15 (16.9%)	104 (48.8%)

**Table 2 ijerph-19-04267-t002:** Confirmatory factor analysis and reliability analysis.

Construct and Scale Items	*λ*	AVE	C.R.	*α*
**Loyalty to sport**		0.667	0.856	0.817
At a time the government quarantine policy in sport has been relaxed,				
I will continue to like the sports I participate in.	0.880			
I get information related to sports participation through various media.	0.721			
Participation in sports accounts for a large part of my leisure activities.	0.840			
**Behavioral intention for participation in sport**		0.756	0.902	0.869
At a time the government quarantine policy in sport has been relaxed,				
If I have a chance in the future, I will continue to participate in sports.	0.916			
I will recommend my friends to participate in sports in the future.	0.812			
I will purchase sports equipment to participate in sports in the future.	0.876			
**Infection anxiety from others**		0.791	0.938	0.889
When participating in sports,				
I’m anxious if others don’t follow cough etiquette.	0.868			
I’m anxious about whether my mask is properly worn.	0.791			
I feel uneasy when I contact other people without wearing a mask.	0.954			
I feel uneasy when talking to others who are not wearing masks while I am wearing a mask.	0.936			
**Risk perception of COVID-19 (sensitivity)**		0.742	0.896	0.862
I have a risk of being infected with respiratory infections.	0.778			
I feel anxious that it will be transmitted.	0.861			
I feel that there is a high possibility of being infected with respiratory infectious diseases.	0.939			
**Risk perception of COVID-19 (seriousness)**		0.644	0.843	0.803
If I am infected with COVID-19, even my family can be infected.	0.847			
My daily life becomes uncomfortable if I am infected with COVID-19.	0.870			
If I am infected with COVID-19, I should receive tough treatment.	0.677			

CMIN = 223.840, DF = 94, CMIN/DF = 2.381, NFI = 0.938, CFI = 0.963, RMSEA = 0.068.

**Table 3 ijerph-19-04267-t003:** Results of one-way multivariate analysis of variance.

Dependent Variables	*df*	*F*	*p*	*η^2^*
Loyalty to sport	1	95.958	0.000 ***	0.242
Behavioral intention for participation in sport	1	26.290	0.000 ***	0.081
Infection anxiety from others	1	72.920	0.000 ***	0.196
Risk perception of COVID-19 (sensitivity)	1	14.338	0.000 ***	0.046
Risk perception of COVID-19 (seriousness)	1	0.039	0.844	0.000

*** *p* < 0.001.

**Table 4 ijerph-19-04267-t004:** Mean scores of variables between groups.

	1	2	3	4	5
Unvaccinated	2.727	3.479	2.424	**3.494**	3.489
Vaccinated	**3.789**	**4.056**	**3.548**	3.014	3.466

1 = Loyalty to sport, 2 = Behavioral intention for participation in sport, 3 = Infection anxiety from others, 4 = Risk perception of COVID-19 (sensitivity), 5 = Risk perception of COVID-19 (seriousness). Statistically significant higher mean scores between groups are reported in bold.

## Data Availability

Not applicable.

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
