# Peer review of "A Gap between Relaxation of Government Quarantine Policy and Perceptions of COVID-19 among the General Public in Sports: Focusing on Vaccination Status"

_ijerph, 2022, doi:10.3390/ijerph19074267_

Round 1

Reviewer 1 Report

Preliminary comment

The ethical committee (code) is not mentioned in the document. Add this information to the bottom of the document, as part of the IJERPH rules.

ABSTRACT

Lines 17-18: add the nationality and context (e.g., students, active people) of the population

Lines 18-21: a validated instrument (survey) was employed to obtain the data?

Lines 21-23: results should be clearly improved. The current report does not provide an answer to the main research question. Moreover, statistical values are missing.

Lines 23-25: the conclusion is vague and does not supported on the results.

INTRODUCTION

Lines 29-31: The third pandemic in history? Or pandemic of a coronavirus? Check this data: https://www.euro.who.int/en/health-topics/communicable-diseases/influenza/pandemic-influenza/past-pandemics

Line 70: A related work is missing before introducing the objective of the study. The vast majority of the introduction is the presentation of the COVID impact, but few are related to the main topic of the research. Rationale must be strengthened. Use references like these to build a related work paragraph that justifies the contribution:

Mutz, M., & Gerke, M. (2021). Sport and exercise in times of self-quarantine: How Germans changed their behaviour at the beginning of the Covid-19 pandemic. International Review for the Sociology of Sport56(3), 305-316.

de Abreu, J. M., de Souza, R. A., Viana-Meireles, L. G., Landeira-Fernandez, J., & Filgueiras, A. (2022). Effects of physical activity and exercise on well-being in the context of the Covid-19 pandemic. PloS one17(1), e0260465.

Khosravi, M. (2020). COVID-19 quarantine: Two-way interaction between physical activity and mental health. European Journal of Translational Myology30(4).

METHODS

As a cross-sectional study, I would like to recommend following the STROBE Guidelines.

Before data collection and procedure: Present key elements of study design early in the paper and describe the setting, locations, and relevant dates, including periods of recruitment, exposure, follow-up, and data collection

Line 131: add a priori sample size estimation (using G*Power or similar)

Lines 147-149: add details about the criteria for including or excluding the answers.

Lines 165-179: for each survey, add the way of measurement, and describe the main outcomes extracted.

Line 179: a section of statistical procedures is missing. Describe all statistical methods, including those used to control for confounding. Describe any methods used to examine subgroups and interactions. Explain how missing data were addressed.

RESULTS

Section 3.2. descriptive statistics are missing. Add tables or graphs to represent the descriptive statistics obtained per group.

Table 3: present the pairwise comparisons, with p and cohen’s d information.

DISCUSSION

Lines 216-232: reduce the background, and summarise key results with reference to study objectives

Most of the discussion must be strengthened with references to support the hypothesis and consider the results from similar studies, and other relevant evidence.

Examples of articles to support some discussions:

Burke, P. F., Masters, D., & Massey, G. (2021). Enablers and barriers to COVID-19 vaccine uptake: An international study of perceptions and intentions. Vaccine39(36), 5116-5128.

Lin, C., Tu, P., & Beitsch, L. M. (2020). Confidence and receptivity for COVID-19 vaccines: a rapid systematic review. Vaccines9(1), 16.

Karlsson, L. C., Soveri, A., Lewandowsky, S., Karlsson, L., Karlsson, H., Nolvi, S., ... & Antfolk, J. (2021). Fearing the disease or the vaccine: The case of COVID-19. Personality and individual differences172, 110590.

Line 285: add a paragraph with study limitations, future research, and practical implications.

Author Response

Reviewer 1

The ethical committee (code) is not mentioned in the document. Add this information to the bottom of the document, as part of the IJERPH rules. à We totally understand your concern. In terms of the approval code provided by the competent Ethics Committee that you mentioned, this study is a research in social science which has never collected sensitive personal information from survey participants (a reason for exemption from IRB review). Additionally, all participants were informed about the detailed information of this study and their rights as a survey respondent. As a result, this study did not have the IRB review process.

ABSTRACT

  • Lines 17-18: add the nationality and context (e.g., students, active people) of the population →  As you mentioned, the Abstract has been revised.
  • Lines 18-21: a validated instrument (survey) was employed to obtain the data? → Yes, we have employed validated instruments that have been utilized from previous studies. In addition, the information of the instruments has been reported in the section of 2. Instruments.
  • Lines 21-23: results should be clearly improved. The current report does not provide an answer to the main research question. Moreover, statistical values are missing. → As you mentioned, the Abstract has been revised (main research question and statistical values).
  • Lines 23-25: the conclusion is vague and does not supported on the results. → As you mentioned, the Abstract has been revised.

INTRODUCTION

  • Lines 29-31: The third pandemic in history? Or pandemic of a coronavirus? Check this data: https://www.euro.who.int/en/health-topics/communicable-diseases/influenza/pandemic-influenza/past-pandemics → As you mentioned, the sentence has been revised.
  • Line 70: A related work is missing before introducing the objective of the study. The vast majority of the introduction is the presentation of the COVID impact, but few are related to the main topic of the research. Rationale must be strengthened. Use references like these to build a related work paragraph that justifies the contribution: → We totally understand your comment. The paragraph has been revised with the additional references.

Mutz, M., & Gerke, M. (2021). Sport and exercise in times of self-quarantine: How Germans changed their behaviour at the beginning of the Covid-19 pandemic. International Review for the Sociology of Sport56(3), 305-316.

de Abreu, J. M., de Souza, R. A., Viana-Meireles, L. G., Landeira-Fernandez, J., & Filgueiras, A. (2022). Effects of physical activity and exercise on well-being in the context of the Covid-19 pandemic. PloS one17(1), e0260465.

Khosravi, M. (2020). COVID-19 quarantine: Two-way interaction between physical activity and mental health. European Journal of Translational Myology30(4).

METHODS

  • As a cross-sectional study, I would like to recommend following the STROBE Guidelines. → We absolutely understand your concern. The Strengthening the Reporting of Observational Studies in Epidemiology (STROBE) Statement is a guideline for reporting observational studies. However, STROBE is not suitable for this study which is a survey research in social science. Through this revision process, we have strengthened the section of Materials and Methods based on reviewers’ comments.
  • Before data collection and procedure: Present key elements of study design early in the paper and describe the setting, locations, and relevant dates, including periods of recruitment, exposure, follow-up, and data collection → Based on your comment, the paragraph has been revised for better understanding.
  • Line 131: add a priori sample size estimation (using G*Power or similar) → Based on your comment, we added the result of G*Power.
  • Lines 147-149: add details about the criteria for including or excluding the answers. → Based on your comment, the paragraph has been revised for better explanation.
  • Lines 165-179: for each survey, add the way of measurement, and describe the main outcomes extracted. → Based on your comment, more detailed information (content validity and readability) has been added.
  • Line 179: a section of statistical procedures is missing. Describe all statistical methods, including those used to control for confounding. Describe any methods used to examine subgroups and interactions. Explain how missing data were addressed. → We understand your concern. This study has offered the statistical procedures in the section of Results. In addition, detailed information regarding independent and dependent variables was described in the section of 2. Materials and Methods. Last, incomplete questionnaires (missing data) have been excluded in this study.

RESULTS

  • Section 3.2. descriptive statistics are missing. Add tables or graphs to represent the descriptive statistics obtained per group. → The table 1 has included the descriptive statistics obtained per group.
  • Table 3: present the pairwise comparisons, with p and cohen’s d information. → We totally understand your concerns. However, this research has conducted a comparative study comparing two groups. Thus, the pairwise comparisons are not meaningful in that the MANOVA has already found the differences between two groups. In addition, the purpose of this study has not been to find how large the effect is (Cohen’s d), but to find whether the effect exists (MANOVA).

DISCUSSION

  • Lines 216-232: reduce the background, and summarize key results with reference to study objectives → Based on your comment, the paragraph has been reduced.
  • Most of the discussion must be strengthened with references to support the hypothesis and consider the results from similar studies, and other relevant evidence. → We totally understand your comment. The paragraph has been revised with additional references.

Examples of articles to support some discussions:

Burke, P. F., Masters, D., & Massey, G. (2021). Enablers and barriers to COVID-19 vaccine uptake: An international study of perceptions and intentions. Vaccine39(36), 5116-5128.

Lin, C., Tu, P., & Beitsch, L. M. (2020). Confidence and receptivity for COVID-19 vaccines: a rapid systematic review. Vaccines9(1), 16.

Karlsson, L. C., Soveri, A., Lewandowsky, S., Karlsson, L., Karlsson, H., Nolvi, S., ... & Antfolk, J. (2021). Fearing the disease or the vaccine: The case of COVID-19. Personality and individual differences172, 110590.

  • Line 285: add a paragraph with study limitations, future research, and practical implications. → Based on your comment, we improved the paragraph in Conclusions and limitations by adding study limitations, future research, and practical implications.

Reviewer 2 Report

Congratulations on the work done.

In detail, my comments are:

 Abstract:

It is important to highlight significant results in the abstract.

Introduction:

Line 35: What exactly do you mean by: “…and many more infections and deaths are not statistically significant”. I think it is necessary to complete this information with objective data in order to speak of “not statistically significant”.

Line 41: “…the government prepared for step-by-step daily recovery to coexist with COVID-19 by changing perception and the quarantine system”. Authors should put some examples.

Line 44: The authors should explain what it is: “relaxed quarantine system” and “high-intensity social distancing”.

Line 46: Although we currently know what “Omicron” is, it should be clear what this variant is, as well as which variants of the virus have developed over these two years.

The authors speak of different variables throughout the text, for example, "anxiety, happiness, mental health, physical health, life satisfaction" but these variables are not defined or clarified. The introduction should focus on the variables analyzed in this investigation.

If the objective indicated at the end of the introduction also points to future strategies in the face of the pandemic situation, it would be important to know which strategies have been used in previous studies so far. The study objectives must be in accordance with what is presented in the results.

There is also no review of what happens in the variables that are intended to be studied in subjects who are vaccinated and who are not vaccinated (e.g., anxiety).

The introduction must be rewritten and improved, indicating consistency between the information presented and the concrete variables studied in this investigation.

Materials and Methods

It must be completed and organized (e.g., participants, procedure,…), with clear information (e.g., control of the criteria considered in the study).

Tables should present their notes similarly throughout the document.

The expression “this study modified the instruments developed” should be modified. Authors should use more accurate wording when building or adapting new research instruments. Psychometric information from the original instruments must be added.

There must be clear indication and justification of the data analysis used, as well as compliance with statistical assumptions.

Discussion

It should present more studies that support the variables analyzed in the results. In fact, psychological and behavioral characteristics are widely studied in various contexts and populations.

Author Response

Reviewer 2

Abstract:

  • It is important to highlight significant results in the abstract. → As you mentioned, the Abstract has been revised.

Introduction:

  • Line 35: What exactly do you mean by: “…and many more infections and deaths are not statistically significant”. I think it is necessary to complete this information with objective data in order to speak of “not statistically significant”. → As you mentioned, the sentence has been revised to meet the sentence intention.
  • Line 41: “…the government prepared for step-by-step daily recovery to coexist with COVID-19 by changing perception and the quarantine system”. Authors should put some examples. → As you mentioned, the sentence has been revised by adding some examples.
  • Line 44: The authors should explain what it is: “relaxed quarantine system” and “high-intensity social distancing”. → As you mentioned, the sentence has been revised with additional explanations.
  • Line 46: Although we currently know what “Omicron” is, it should be clear what this variant is, as well as which variants of the virus have developed over these two years. → As you mentioned, we have added a brief explanation in that this research is not a medical research.
  • The authors speak of different variables throughout the text, for example, "anxiety, happiness, mental health, physical health, life satisfaction" but these variables are not defined or clarified. The introduction should focus on the variables analyzed in this investigation. → We absolutely agree with this comment. However, the terms you mentioned are not the main variables in this study. Thus, the terms will be fully understood in terms of common sense by readers. In case of dependent variables, Loyalty, Behavioral intention for participation, Infection anxiety, and Risk perception, this study has provided the definitions and explanations in the section of Introduction.
  • If the objective indicated at the end of the introduction also points to future strategies in the face of the pandemic situation, it would be important to know which strategies have been used in previous studies so far. The study objectives must be in accordance with what is presented in the results. → We absolutely agree with this comment. The future strategies have been stated as practical contributions of this study, and the study objectives have been stated based on the results of this study throughout this manuscript.
  • There is also no review of what happens in the variables that are intended to be studied in subjects who are vaccinated and who are not vaccinated (e.g., anxiety). → We totally understand your concern. Thus, the section of Introduction has been revised by adding reviews of previous studies regarding the main variables of this study.
  • The introduction must be rewritten and improved, indicating consistency between the information presented and the concrete variables studied in this investigation. → We totally understand your review comments. Thus, the section of Introduction has been improved for the consistency.

Materials and Methods

  • It must be completed and organized (e.g., participants, procedure,…), with clear information (e.g., control of the criteria considered in the study). → Based on your comment, the paragraph has been revised for better understanding.
  • Tables should present their notes similarly throughout the document. → Based on your comment, all tables have been checked.
  • The expression “this study modified the instruments developed” should be modified. Authors should use more accurate wording when building or adapting new research instruments. Psychometric information from the original instruments must be added. → Based on your comment, more detailed information (content validity and readability) has been added. In addition, psychometric information of the original instruments has been added.
  • There must be clear indication and justification of the data analysis used, as well as compliance with statistical assumptions. → We understand your concern. This study has described the statistical analyses in the section of Results. Also, detailed information regarding independent and dependent variables was described in the section of 2. Materials and Methods. Last, information about the statistical assumption has been added.

Discussion

  • It should present more studies that support the variables analyzed in the results. In fact, psychological and behavioral characteristics are widely studied in various contexts and populations. → We totally understand your comment. The paragraph has been revised with additional references.

Reviewer 3 Report

The topic is interesting and provide some useful information regarding the appropriateness of quarantine policies on sports participation and respiratory infection perception according to vaccination status.

The methodology and the clearness of presentation of the work are quite good.  Nevertheless, I have a fundamental question that could affect the appropriateness of study design and the results of the study: In the introduction it is not specified if there are or not different restriction between vaccinated and unvaccinated regarding the access to gyms, swimming pools, or other sport facilities. If different rules are applied, these could have influenced the results of this study and need attention.

In addition, I think that it is important to know if sex and age influence sports participation and respiratory infection perception.

Author Response

Reviewer 3

  • The methodology and the clearness of presentation of the work are quite good. Nevertheless, I have a fundamental question that could affect the appropriateness of study design and the results of the study: In the introduction it is not specified if there are or not different restriction between vaccinated and unvaccinated regarding the access to gyms, swimming pools, or other sport facilities. If different rules are applied, these could have influenced the results of this study and need attention. → We totally understand your concern regarding restrictions to sport facilities. As described in the manuscript, this study was implemented at a time the government quarantine policy has been relaxed, and the government policies were equally applied regardless of vaccination status.
  • In addition, I think that it is important to know if sex and age influence sports participation and respiratory infection perception. → We totally understand your comment. This study was not able to apply all potential factors (e.g., age) which could be significant on sports participation and respiratory infection perception. Thus, this study has already described the research limitation in the section of Conclusions and limitations. Additionally, we have strengthened the paragraph based on your comment.

Reviewer 4 Report

I have reviewed the "A gap between relaxation of government quarantine policy and perceptions of COVID-19 among the general public in sports: Focusing on vaccination status" document. Before continuing your processing, it is necessary to pay attention to the following:

- Check the wording throughout the document, do not write in the first person, Line 15...
- Check the wording time throughout the document.
- I suggest deleting subheadings 1.1, 1.2 and 1.3. and move these paragraphs before the last paragraph of the introduction.
- In the first paragraph of the conclusions it says absolutely nothing.
- The results should respond to the objectives of your research, now they are not clear.
- In the first paragraph of the results you can see part of the methodology.
- In the paragraph of the results I do not observe anything about the relevance of the policies.
- The discussion should synthesize more, generate a paragraph for the specific objectives of your research.
- It would be good to describe the policies of the study area, in order to analyze the relaxation.

Author Response

Reviewer 4

  • Check the wording throughout the document, do not write in the first person, Line 15... → As you mentioned, the wording has been revised.
  • Check the wording time throughout the document. → The meaning of “wording time” is not clear.
  • I suggest deleting subheadings 1.1, 1.2 and 1.3. and move these paragraphs before the last paragraph of the introduction. → As you mentioned, we have deleted the subheadings and moved the paragraphs.
  • In the first paragraph of the conclusions it says absolutely nothing. → Based on your comments, the section of Conclusions has been revised thoroughly.
  • The results should respond to the objectives of your research, now they are not clear. → Results should cover all results of data analyses of this study. The first part of the results has described scale validity and reliability analyses. After that, in the section of 2 Multivariate Analysis of Variance (MANOVA) has described the objectives of this comparative research by showing differences of dependent variables between two groups.
  • In the first paragraph of the results you can see part of the methodology. → We totally understand your concern. However, the first paragraph has described the results of validity and reliability analyses for the data.
  • In the paragraph of the results I do not observe anything about the relevance of the policies. → We absolutely understand your comment. Prior to data collection, all survey respondents were informed about the purpose of this study via informed consent. However, the detailed information informed to the participants have not been fully described in this manuscript. This study attempted to analyze people's thoughts on relaxation of government quarantine policy in sport through their loyalty and behavioral intention in sport. It was to understand the psychology of people at psychological risk of viral infection in a situation where restrictions on sports facilities are being lifted. Thus, we added a specific explanation based on your comments.
  • The discussion should synthesize more, generate a paragraph for the specific objectives of your research. → Based on your comment, the section of Discussion has been revised.
  • It would be good to describe the policies of the study area, in order to analyze the relaxation. → Based on your comment, the section of Discussion has been strengthened.

Round 2

Reviewer 1 Report

Can be accepted in the current form

Author Response

Reviewer 1

Thank you for your careful review comments.

Reviewer 2 Report

Congratulations on the work done.

Author Response

Reviewer 2

Thank you for your careful review comments.

Reviewer 3 Report

In the reply the Authors claimed that "this study was implemented at a time the government quarantine policy has been relaxed, and the government policies were equally applied regardless of vaccination status". Since in other countries different rules have been applied during the post quarantine time, this aspect need to be underlined. In line 47 I suggest to add 'without any difference between vaccinated and unvaccinated.

As I have already reported I think that it is important to know if sex and age influence sports participation and respiratory infection perception and I think you have all the elements (age classes and age) to value this aspects.

I should be able to add in ANOVA table with the results by sex and age classes.

Please report the inference results in table 4.

Author Response

Reviewer 3

In the reply the Authors claimed that "this study was implemented at a time the government quarantine policy has been relaxed, and the government policies were equally applied regardless of vaccination status". Since in other countries different rules have been applied during the post quarantine time, this aspect need to be underlined. In line 47 I suggest to add 'without any difference between vaccinated and unvaccinated. → Based on your comment, the sentence has been revised.

As I have already reported I think that it is important to know if sex and age influence sports participation and respiratory infection perception and I think you have all the elements (age classes and age) to value these aspects. I should be able to add in ANOVA table with the results by sex and age classes. Please report the inference results in table 4. → We agree with your comment that the factors (i.e., sex and age) might be significant on sports participation and respiratory infection perception. However, as I mentioned, the purpose of this study was to find a gap between relaxation of government quarantine policy and perceptions of COVID-19 among the general public in sports depending on vaccination status. If this study applies these factors (i.e., sex and age) as significant variables and conducts additional statistical analyses (i.e., ANOVA), the purpose, topic, and design of this study should be totally changed. That is why this study has not applied the variables. Thus, this study has described what you mentioned in the section of 5. Conclusions and limitations (as a research limitation).

Reviewer 4 Report

The manuscript has improved. 
But in the conclusions section it is necessary to separate at least two paragraphs. And include a third on the limitations and future studies based on its results.

Author Response

Reviewer 4

But in the conclusions section it is necessary to separate at least two paragraphs. And include a third on the limitations and future studies based on its results. → Based on your comment, the paragraph has been separated and improved.
